# Accommodation Strategies in Education: Exploring Jesuit Textbook Publication in Modern Shanghai

Wei Mo [1,2,3]

1   Department of Foreign Studies, Shanghai Maritime University, Shanghai 201306, China;
    mowei@shmtu.edu.cn
2   History Department, Fudan University, Shanghai 200433, China
3   Department of Theology, Uppsala University, 75105 Uppsala, Sweden

**Abstract:** This study examines the Jesuits' significant contributions to China's modernization through their involvement in textbook publication. It analyzes a recently discovered historical *Catalogus* of 1934 written in Latin to uncover insights. Specifically, it explores how the Jesuit T'ou-sè-wè Press in Shanghai responded to changes in education, society, and the interplay of religious and political forces. By investigating the Jesuits' strategic adaptations in their educational and cultural pursuits, revised compilation and publishing methods, and proactive engagement with the dynamic environment, this study aims to provide a deeper understanding of their approach. Additionally, it investigates the nuanced accommodation strategies employed by the T'ou-sè-wè Press to navigate the complex challenges of the era.

**Keywords:** the restored Jesuits; T'ou-sè-wè Press; textbook publication; the Jesuit accommodation policy; modern Shanghai





## 1. Introduction

Since the 1870s, the Jesuit Zi-ka-wei 徐家匯 compound has transformed into a prominent center for publishing, playing a vital role in science, education, and culture in modern Shanghai. Originally a religious and charitable enclave, this triangular area now contributes to global knowledge dissemination. Through comprehensive publications of monographs and periodicals, Zi-ka-wei integrates regional knowledge into the broader fabric of global civilization. At the southern end of Zi-ka-wei, the T'ou-sè-wè Press 土山灣印書館 serves as an indispensable bastion of knowledge.

This study examines the *Catalogus Librorum Lingua Sinica Scriptorum of Typographia Missions Catholicæ* in Orphanotrophio T'ou-sè-wè (referred to as the *Catalogus*), compiled in 1934, as a primary reference point for investigating the Jesuits' involvement and strategies in textbook publication during their Shanghai mission. It sheds light on the historical continuity and revitalization of the Jesuits' "accommodation policy", revealing their role in knowledge dissemination and cultural identity preservation amidst China's modernization.

Four key areas are the major focus: the naming and linguistic strategies in the *Catalogus*, the selection of representative textbooks, the compilation and publishing strategies employed by the Jesuits in Zi-ka-wei, and the transformative impact of Jesuit textbook publications on educational concepts. It uncovers the multifaceted approach adopted by the Jesuits, their proactive response to societal transformations, and their influential role in shaping educational practices within China's intellectual community. This research provides valuable insights into the Jesuits' mission, highlighting their contribution to knowledge and cultural development.

**2. Literature Review**

T'ou-sè-wè should be acknowledged as a comprehensive institution housing various workshops, with the T'ou-sè-wè Press and Atelier standing out as particularly well-studied approaches (Ma 2016). Concerning the scholarly works published by the T'ou-sè-wè Press, the scholarships adopt a case-oriented approach that delves into individual works and authors. This method allows for comprehensive analyses of both focused and scattered aspects, ultimately contributing to a nuanced understanding of the subjects being investigated.

The focused analyses concentrate on individual works and specific authors. Notable examples include studies on Henri Dorè 禄是遒's *Recherches sur les Superstitions en Chine* 中國迷信研究 (Li 2012, pp. 313–29), Angelo Zottoli 晁德蒞's *Cursus Litteraturæ Sinicæ* 中國文化教程 (De Caro 2022; Liu 2015; Ahn and Moon 2020; Williams 2015; Si 2016), and Aloysius Gaillard 方殿華's *Nankin d'Alors et d'Aujourd'hui* 金陵古今 (Pelliot 1903). These comprehensive and profound case studies are situated within a broader temporal perspective, allowing for a balanced consideration of both focal and scattered aspects. By contextualizing them within a wider framework, a deeper exploration can be undertaken within the historical and cultural context.

The scattered mode of analysis typically focuses on a series of branded publications, presenting a macro-systematized framework. This approach often involves specific collections such as the Chinese Christian Texts from the Zi-ka-wei Library (Standaert 1996, 2013; Tao 2017; Shanghai Library 2020) and branded book series like *Variétés sinologiques* (Wang 2016). It facilitates the establishment of reference points for defining the literary significance of specific works. Employing a two-fold strategy, this dynamic approach vividly constructs the knowledge production and dissemination landscape of modern Zi-ka-wei.

It should be noted here that when exploring textbook publications, the attention has predominantly been directed towards large private publishing institutions with dedicated textbook editing departments, such as the Commercial Press 商務印書館 (Culp 2019; Zhang 2021; Wang 2017) and the Kai Ming Bookstore 開明書店 (Yu 2019). This exploration also extends to the Educational Association of China 益智書會 (Bennett 1967; Zhang 2011, 2020) established by Protestant missionaries in China. Regrettably, the Jesuit-led T'ou-sè-wè Press has received scant attention in this regard (Zou 2010).

**3. The *Catalogus* Librorum Lingua Sinica Scriptorum of 1934**

*3.1. The Content*

The *Catalogus* distinguishes itself from the typical specialized book catalogs produced by the T'ou-sè-wè Press. While the Press regularly released catalogs listing works published each year, such as the *Catalogue des Ouvrages Européens* for Western language publications (Mo 2022) and the *Catalog of Scriptures* specifically compiled for Jesuit "Chinese-Western Books" during the Ming and Qing dynasties (Golvers 2012–2015), the comprehensive Latin *Catalogus* provides a panoramic overview of T'ou-sè-wè Press' publications each year, spanning approximately 140 pages. The abbreviated *Catalogus major 1934* is noted on the spine. Written in authoritative Latin, in line with the Catholic context, the *Catalogus* features detailed sections presenting book titles in Romanized Wu dialect phonetics, followed by Chinese characters, and supplemented with Latin information including updates on editions, printing techniques, page specifications, and selling prices.

The *Catalogus* is divided into two main sections: Libri Religiosi (Religious Books) and Libri Profani (Secular Books). The Libri Religiosi section encompasses various branches such as Scriptura Sacra, Hagiographia, Apologetica, Doctrinales, Catechistica, Devotiones, Lectiones spirituales, Meditationes, Ritus, Calendarla, Musica, Condices and Folia, totaling 13 branches. Each book within this section is assigned a unique number, resulting in a total of 635 items, with very few missing numbers. The Libri Religiosi section accounts for over seventy percent of the *Catalogus*.

The Libri Profani section is also organized into 13 branches, covering subjects such as De Stylo Sinico, Calligraphia, Historia, Biographiæ, Geographia, Philosophia, Mathesis,

Scientiæ Physicæ et Naturales, Medicina, Ars Delineandi, Periodica, Parvæ Narrationes Morales, and Varia. Each book in this section is likewise assigned a unique corresponding number, many of which belong to multi-volume sets.

The *Catalogus* was aligned with the Jesuit framework of the Plan Scientifique du Kiang-nan, as described by Mo (2021), which involved extracting publications with instructional qualities from the Libri Profani section and categorizing them by language and science. This approach sought to emphasize the Jesuit's multidimensional efforts in localization, inculturation, and networking (Lee 2004). These dimensions reflect the proactive engagement of the Jesuits in the modernization process in China within the historical context. The practices of knowledge circulation and the cultivation of cultural identity through these publications have become integral elements of Jesuit scientific and educational endeavors in modern Shanghai, playing a significant role in shaping the intellectual landscape and fostering a sense of cultural belonging within the community.

### 3.2. The Press: The First Combined Entity as T'ou-sè-wè Cimutang

The cover of the *Catalogus* prominently displays the inscription "Typographia Missions Catholicæ in Orphanotrophio T'ou-sè-wè", identifying the publishing institution as the T'ou-sè-wè Cimutang Press 土山灣慈母堂. Historically, T'ou-sè-wè and Cimutang are referred to separately, and rarely used together. T'ou-sè-wè was associated with French works printed using Western techniques, while Cimutang editions were Chinese-style reproductions of works by Jesuit missionaries and Chinese–Western books from the Ming and Qing dynasties, employing traditional Chinese methods. The extensive collection of Cimutang publications represents the rich history of the Jesuits in China, spanning from the late 16th century to the mid-20th century. This history is not simply a linear succession of events, but rather a complex tapestry of intertwined narratives.

The 1934 *Catalogus* explicitly identified the combined entity as T'ou-sè-wè Cimutang, bridging the Jesuit accommodation efforts across centuries. In the case of the textbooks put out by the T'ou-sè-wè Cimutang Press, the Jesuits exemplified Catholic localization progress in the 20th century, defended a counter-cultural perception and aligned with a linear historical narrative. By adopting this combined entity as T'ou-sè-wè Cimutang in the *Catalogus*, the Jesuits actively engaged in ongoing dialogue on political and religious tensions within the restored Jesuit mission in modern Shanghai.

### 3.3. The Language Pattern: Wu Dialect-Latin

The languages chosen for the *Catalogus*, Latin and the Romanized spelling of the Wu dialect, deviated from the French commonly used in the Zi-ka-wei mission compound. This deliberate decision by the Jesuits aimed to strike a balance between universality and localization.

The language choice may have also involved political considerations in language planning. The "generation of giants" (Dunne 1962) initiated the romanization of the Church dialect as part of the accommodation policy, laying the groundwork for this modern movement. In the 1930s, there was a growing recognition of dialects in China, and national intellectuals proposed the Dialect Romanization Movement to modernize the language (Zhan 2016). The Jesuits, understanding the persuasive power of dialects among their primarily illiterate followers, played a significant role in this movement. Despite their classical education, the Jesuits recognized the importance of engaging with dialects to effectively instruct and inspire their audience.

Through their extensive network of correspondences, the Jesuits' efforts resonated with the global language reformation. The practice of Romanizing the Wu dialect began in Zi-ka-wei and spread to other areas where the Jesuits were active, such as Zo-sè 佘山, Lou-ka-wè 卢家湾, and Loh-ka-pang 陆家浜. This integration of seemingly contrasting elements, rooted in the local context yet emanating a universal perspective, served as a catalyst for advancements in language theory. Both Jesuit scholars and other linguists benefited significantly from the contributions of the T'ou-sè-wè Cimutang Press.

In summary, the *Catalogus* showcases the accommodation strategy of T'ou-sè-wè, blending local social contexts with universal academic initiatives. The publisher's dynamic self-naming approach and the use of multiple languages dissolve language–political intentions within the global academic system. Understanding the environmental factors that shaped modern textbooks in Zi-ka-wei is crucial.

## 4. Shaping the Environment and Purpose

In the modern era, the compilation of textbooks in Zi-ka-wei goes beyond a simple adherence to educational principles and encompasses a comprehensive approach. This approach takes into consideration the institutional environment, including changes in the education system, criticisms of Catholic education, and the influence of religious elites. Recognizing the multidimensional nature of education, the textbook publishing strategy in modern Zi-ka-wei involves negotiations and compromises across various dimensions. Ultimately, this approach aligns with the holistic education philosophy of *Cura Personalis* (Kolvenbach 2007), emphasizing the care and respect for every individual. It reflects the cultural and educational advancements observed in Zi-ka-wei.

### 4.1. The Surrounding Factors

Zi-ka-wei's modern educational endeavors in textbook compilation and publishing exemplify a multidimensional approach that considers institutional dynamics, challenges, and the interplay between Christianity and educational missions.

Firstly, the implementation of the 1922 National Education Reform, which emphasized regional autonomy and flexibility, liberated education from traditional constraints (Keenan 1974). This reform propelled Zi-ka-wei's educational progress by allowing for innovation and adaptability.

Secondly, within the complex context of colonial modernity (Laamann 2020, pp. 106–7), the relationship between Christianity and educational missions has significantly influenced Zi-ka-wei's educational landscape. While Protestant groups sought dominance in educational discourse (Lutz 1971), the Catholic Church faced challenges and diminishing influence (Tiedemann 2010, pp. 516–25). However, amidst critiques of privilege, the Church sought collaboration with the French academic community, opening a realm of possibilities. Zi-ka-wei's multidimensional approach provided an ideal platform for navigating these dynamics.

Lastly, the Jesuits' philosophy of *Cura Personalis*, which focuses on the holistic development of individuals, aligns with Zi-ka-wei's comprehensive educational system. Despite their prominence in university education, the Jesuits have long recognized the importance of educating children and those without formal schooling (O'Malley 2000b). Zi-ka-wei's comprehensive approach provides an opportunity for the harmonious synthesis of ancient and modern educational philosophies.

Overall, Zi-ka-wei's modern educational initiatives in textbook compilation and publishing, driven by institutional adaptability, the complexities of colonial modernity, and the Jesuit philosophy of *Cura Personalis*, have established a comprehensive and multidimensional educational system. This system celebrates autonomy, challenges traditional frameworks, nurtures individuals' Christian faith, and serves as a timeless exemplar for similar educational contexts.

### 4.2. The Paradigmatic Effect of the Catalogus
4.2.1. Centered on Zi-ka-wei

In modern Zi-ka-wei, educational services are provided through a diverse range of schools. These facilities, established over several decades, cater to both religious and secular education. They include seminaries and a theology college for clergy formation, as well as universities, girls' schools, vocational education institutions, and special education facilities. Each institution operates independently while demonstrating a commitment to public service and meeting diverse needs.

As a supporting institution, T'ou-sè-wè is responsible for publishing and printing textbooks. These textbooks can be categorized into two main groups: those designed for church services, including materials for missionary works, catechism, and spiritual writings, and those aligned with the national education guidelines, used by both secular and church-affiliated schools. The *Catalogus*, with its sections for religious and secular books, effectively complements the educational needs of Zi-ka-wei's institutions.

### 4.2.2. The Shared Philosophy

The establishment of the subject in modern Zi-ka-wei reflects the shared philosophy between the 1922 National Education Reform and the Jesuit Noveau Ratio Stadium (Pavur 2005, pp. 145–70). This approach recognizes the importance of adapting religious educational philosophy to the local context. while also driving universality. In line with the Jesuit education philosophy found in the Ratio, there is a focus on academic specialization, particularly in science and theology, while also acknowledging the significance of Christian conduct and rhetoric (O'Malley 2000a). Modern Zi-ka-wei, such as St. Ignatius College, demonstrates this philosophy by making accommodation adjustments to its subject structure, exemplified by retaining the Course of *Ethicæ* alongside national education's *Civics*.

The subject structure includes a range of philosophical works, such as *Elementa Philosophiæ* 哲學提綱 by Laurentio Li 李問漁, and *Compendium de historia Philosophiæ* 哲學史綱 by Joseph Siu 徐宗澤. Additionally, a sub-category called "Tracts" focuses on contemporary social issues. The Philosophia category acts as a bridge between national and religious education, allowing for mutual references. For example, *Elementa Philosophiæ Ethica seu Moralis* 哲學史綱 倫理學 covers various topics, encouraging students to analyze them using approaches from both church ethics and the social sciences. This convergence of educational types exemplifies the model Zi-ka-wei's educational undertakings aimed to embody.

### 4.2.3. Unveiling the Controversies

Textbooks provide valuable insights into the intellectual trends of the early 20th century. One notable example was the advocacy of aesthetic education by Cai Yuanpei, the first Minister of Education in the Republican Government, in 1917. Cai argued that aesthetic education could cultivate humanist worldviews, public morality, and emotional and rational development, going so far as to propose it as a potential replacement for religion (Wang 2020). This viewpoint sparked intense debates. In 1925, during the "Reclaim Education Rights Movement", church schools faced challenges amid nationalist fervor, prompting scholars to question the effectiveness of religious moral education and explore alternative approaches.

In the midst of these debates, the T'ou-sè-wè Press published a special issue on Education in June 1926 through its monthly journal, *Renvue Catholique* 聖教雜誌. This special issue covered a wide range of topics, including redefining education based on educational science, exploring the relationships among religions, families, and nations, delving into the history of Catholic education, discussing school organization and textbook publication strategies, adapting to educational reforms, addressing modern language reforms, and examining moral education through religious institutions. These reviews represented the official perspective of the Jesuit Order and contributed to the broader theoretical discussions taking place at the time.

The *Catalogus* took a distinct approach by impartially listing and presenting textbooks that embodied diverse philosophies of education. The aim was to provide a comprehensive overview without religious bias. Some of these texts engaged in speculative discussions that bridged philosophy and theology. For example, Joseph Siu's *Compendium Apologeticæ generalis* explored the origins of humanity, theology, religious studies, and the Catholic Church, starting with René Descartes's famous philosophy of "cogito, ergo sum" as the fundamental principle. The objective was to guide individuals from subjective thinking

to objective understanding, establishing a religious framework centered on reverence for God and reconnecting with one's origins.

Another noteworthy category in the *Catalogus* was Parvæ Narrationes Morales, which incorporated various disciplines such as philosophy, history, and art. This category emphasized the moralizing function of literary works. Works like Kin-piao Fei费金标's *Sanctus Joseph Veteris Testamenti*, *Esther*, and *Machabæi* were presented as dramatic scripts, often performed during Catholic festivals in Zi-ka-wei. These performances vividly demonstrated the effectiveness of moral education and aligned with the literary and artistic paradigm described in Aristotle's *Poetics*.

The goal of the *Catalogus* was to present all categories of education, empowering readers to explore and make their own informed decisions. The concentration and diversity of modern education in Zi-ka-wei facilitated various localized practices. The textbook publications by the T'ou-sè-wè Press played a pivotal role in supporting and promoting the integration of this localized educational experience into broader educational network activities.

## 5. Exploring the Production and Dissemination of Literature and Language Textbooks

The T'ou-sè-wè Press has compiled literature and language textbooks since the late Qing Dynasty, despite lacking a dedicated department. The *Catalogus* showcases 45 Chinese language practice textbooks, along with 8 auxiliary books on calligraphy and copybook exercises. This compilation reflects the interplay between societal demands and individual literacy pursuits.

### *5.1. Three Noteworthy Innovations*

When analyzing the production end, three notable innovations emerge in the writing and compilation of literature and language textbooks: a focus on the personal experiences of Jesuit education in Zi-ka-wei, an awareness of gender equality in education, and a preference for classic Chinese over the vernacular.

### 5.1.1. Drawing from Personal Experiences: A Compilation Approach

The T'ou-sè-wè Press compiled literature and language textbooks with several remarkable features. The personal experiences of the compilers, particularly Jesuit priests from Zi-ka-wei, played a crucial role. Joanne-Baptista Pan 潘谷聲, a prominent figure, compiled eighteen Chinese language textbooks for primary schools. The Bureau Sinologique 光啓社, a sinological research institution in Zi-ka-wei congregated by both the Jesuit sinologists and their Chinese counterparts, contributed twelve textbooks. Pierre Huang 黃伯祿 provided valuable works on letters and contracts, while Andrea Tsiang 蔣邑虛, the principal of St. Ignatius College, contributed essential writing skills resources.

Pan's expertise and contributions shaped the content and pedagogical approach of the T'ou-sè-wè textbooks. The Bureau Sinologique's textbooks catered to different levels of students. Huang's works enriched the textbooks with practical language usage, and Tsiang's resources focused on writing skills.

Recognizing Pan's talents, Tsiang valued his educational experiences and utilized his expertise in compiling the textbooks. Pan's *Liber classicus catholicus ad docendos pueros* 聖教啓蒙課本 and its accompanying reference book became valuable resources.

The T'ou-sè-wè textbooks adapted to educational reforms. Pan reedited volumes and series such as the *Novum manuale linguæ patriæ* 初等小學國文新課本 and the *Compendium grammatica* 文範撮要 to align with changing standards. Supported by the *Synopsis grammaticæ elementaris* 國民學校文法便覽表, the T'ou-sè-wè textbooks provided comprehensive support. Their unique perspectives, extensive content, and pedagogical approach allowed them to thrive in the competitive market for Chinese language textbooks.

5.1.2. Gender Consciousness in Textbook Customization

One notable aspect of the T'ou-sè-wè textbooks was their consideration of gender in customization. In Zi-ka-wei, various institutions for women's education catered to different groups from diverse religious backgrounds and social statuses. The Enlightenment Girls' Middle School, also praised as the Aurora University for Girls, gained a remarkable reputation for its achievements. It attracted wealthier families from non-Catholic backgrounds who chose to enroll their daughters in the school. The curriculum there included a set of textbooks called *Medulla lectionum linguæ patriæ, ad usum scholarum mediarum* 中學國文課本菁華, which were edited by the renowned instructor Zou Tao 鄒弢. Zou was celebrated for his excellent skills in writing short verses, prose, essays, and novels, and he had a significant literary reputation. Originally, the *Medulla lectionum linguæ patriæ* were produced using lithography and were only distributed within Enlightenment Girls' Middle School. However, in 1919, the T'ou-sè-wè Press officially published them, making them available to the public. The textbooks quickly gained significant popularity, even reaching public schools beyond their original intended audience.

Zou Tao's selections included over four hundred pieces, ranging from classical texts like the *Grand Historian of China* and the *History of the Former Han Dynasty*, to prose from the Eight Giants of Tang and Song associated with the Classics movement, as well as contemporary essays by well-known figures such as Zeng Guofan and Liang Qichao. Despite being intended as textbooks for a girls' school, these volumes were purely literary collections without any religious content.

In the preface of the textbooks, Zou Tao explained the original intent behind the selections. He emphasized that purity and elegance were not limited to males alone, and that individuals possessed diverse capacities and faced different difficulties in their studies. Thus, the books they read should cater to their specific needs, stimulating knowledge for some and aiding those struggling with learning difficulties. Zou expressed that it was inappropriate to mix all these different needs together like a great smelting operation.

The T'ou-sè-wè textbooks went beyond the Enlightenment Girls' Middle School. Another notable textbook was *Liber ad discendam linguam patriam* 中學國文讀本, which was specifically designed for female students with specific job preferences in the Normal School affiliated with Notre Dame. The textbook underwent revisions to align with the new education system, dividing chapters into "Texts for Intensive Reading" and "Texts for Brief Reading". The former focused on classical works from the Tang and Song dynasties, while the latter emphasized modern and contemporary writings by authors such as Cai Yuanpei, Hu Shi, Zhu Ziqing, and Ba Jin.

Additionally, the T'ou-sè-wè Press compiled *Exempla epistolarum ad usum Virginum apostolicarum* 聖母院函稿, a manual for teaching nuns at Notre Dame how to write working letters. It contained 79 sample letters covering various situations related to daily pastoral work. In 1934, Bishop Auguste Haouisee 惠濟良 officially entrusted the Presentine Sisters with the responsibility of nurturing virgins, marking a significant historical transformation for this indigenous group. The *Exempla epistolarum ad usum Virginum apostolicarum* provided crucial textual support for the standardization process of the Presentines.

The T'ou-sè-wè textbooks were known for their distinctive feature of addressing gendered perspectives in the literary field. By exploring the characteristics of women's education and proposing classical Chinese concepts, these textbooks aimed to break down the boundaries between religious and secular education, laying a unique foundation for the subsequent development of women's affairs in Zi-ka-wei.

5.1.3. Emphasis on Etiquette

Andrea Tsiang's work, *Budimenta styli epistolaris* 尺牘初桄, was an invaluable resource for individuals seeking to master the art of letter writing. One of the prominent aspects of Tsiang's work was his emphasis on etiquette and cultural refinement. Through detailed guidance and examples, Tsiang highlights the significance of proper etiquette in written communication.

Within his work, Tsiang explores the intricate nuances of etiquette, demonstrating how the choice of words, tone, and structure can convey respect, politeness, and sincerity. He provided practical advice on addressing individuals of different social positions and relationships, emphasizing the importance of using appropriate honorifics and expressions of deference.

Moreover, Tsiang delved into the cultural context surrounding letter writing, recognizing the importance of adhering to established norms and traditions. He explored the use of idioms and literary allusions to enhance the elegance and depth of one's writing, showcasing a profound understanding of Chinese literary heritage.

Similarly, Pierre Huang's work, *Exempla epistolarum inter Missionarios et Mandarinos, notis locupletata* 函牘舉隅, placed a significant focus on the rituals and rules of letter writing in daily life. Huang's meticulous attention to etiquette ensured that his readers are well-versed in the proper conduct of written communication.

Huang's work served as a comprehensive guide for missionaries and late Qing officials, equipping them with the knowledge and skills necessary to navigate the complexities of letter writing within the cultural and social framework of the time. By familiarizing readers with the appropriate forms of address, salutations, and expressions of courtesy, Huang's work enables effective and respectful communication in various contexts.

The combined contributions of Tsiang and Huang provided an invaluable resource for individuals seeking to navigate the intricacies of letter writing with grace and cultural sensitivity. Their works not only provided practical guidance, but also promoted a deeper understanding of the importance of etiquette in fostering meaningful and harmonious communication.

### 5.2. *The Impact of Literature and Language Textbooks*

The T'ou-sè-wè Press' literature and language textbooks have been instrumental in the development of modern Zi-ka-wei. Compiled by national educators, these books reflect the successful localization efforts of the Jesuit mission, incorporating local perspectives. Their translation and dissemination have fostered gender inclusivity while respecting individual developmental trajectories.

### 5.2.1. Translation Value

The original textbooks edited by Pan held significant value in terms of translation. They were translated into French, English, and Spanish languages. The eight-volume French translation of *Novum manuale linguæ patriæ ad usum parvarum scholarum primi gradus* was undertaken by Joseph de Lapparent 孔明道, the editor-in-chief of the *Variétés Sinologiques* series. The English version was translated by John F. Magner 艾若望, a Jesuit from the California Province (Fleming 1987, p. 21). José María Huarte 吳山, a Jesuit from the Castile Province, translated the Spanish version, which interestingly incurred a significantly higher cost compared to the English translation. This multilingual translation of the textbooks exemplifies the mission-oriented approach of the new Jesuits returning to China.

During the mid-19th century, the new Jesuits established nine regional mission centers in China, and collaborated with Chinese Jesuits and missionaries from specific provincial capitals. Jesuit missionaries proficient in French, English, and Spanish from various European and American provinces were involved in the Jesuit mission in the Jiangnan region, encompassing Shanghai, Wuhu 蕪湖, Bengbu 蚌埠, Xuzhou 徐州, Anqing 安慶, and Yangzhou 揚州 (Worcester 2017, p. 164). Zi-ka-wei was chosen as their initial destination for adjustment, preliminary language learning, and cultural assimilation. In contrast to the first-generation Jesuits, who spent years studying in Macau and gaining missionary experience, the new Jesuits had limited time for preparations before coming to China. Their adaptation process in Zi-ka-wei was intense, and the collaborative translation of the multilingual textbooks served to affirm the value of the original books and evaluate their progress in learning Chinese. Furthermore, it provided reliable learning materials for subsequent Jesuits arriving in China, facilitating their integration into the mission.

### 5.2.2. In Tune with Intellectual Innovation

The Jesuit Literature and Language Textbooks demonstrate a responsiveness to intellectual innovation trends. As the education system underwent reformation, numerous publishing institutions emerged, offering a wide array of Chinese language textbooks and fostering a vibrant landscape of educational resources. Zhou Yutong 周予同, an expert in Chinese Classics and chief editor of Chinese textbooks for Commercial Press and Kai Ming Bookstore, advocated for a standard that would enable individuals to proficiently express thoughts and emotions, and narrate facts, using simple classical Chinese. Additionally, he emphasized the importance of developing a basic understanding of Chinese literature and staying abreast of academic changes (Zhou 2019, pp. 415–26).

The Literature and Language Textbooks published by the T'ou-sè-wè Press underwent revisions based on feedback from experts and scholars, with a focus on curriculum time allocation, essay selection, and compilation strategies. The content of these textbooks treated calligraphic training, the origin of Chinese characters, general literature history, and grammar studies as supplementary components separate from the core courses. The selection criteria for articles in the textbooks prioritized argumentative or academic pieces. Extracurricular materials included classical episodic novels, classics, histories, scholars' prose, and a limited selection of poetry and songs.

In terms of format, the Chinese language textbooks adopted a horizontal layout, which facilitated the addition of annotations for foreign texts, departing from the traditional vertical reading habits. Later editions of the textbooks provided concise notes on reference books and chapters, encouraging students to engage in independent research. Notably, comments were omitted to avoid influencing students' judgment, allowing them to form their own opinions.

Overall, the Jesuit Literature and Language Textbooks actively responded to evolving intellectual trends, incorporating new perspectives, and implementing revisions to align with the changing educational landscape.

### 5.2.3. Expanding Sinology: Integrating the Social Sciences

The social contributions of Sinology are evident in various ways. One notable example is the work of Pierre Huang, who received the Prix Stanislas Julien in 1899 for his book *Notions techniques sur la propriété en Chine* 中國產權研究. Huang's book examined the late Qing economic system, the Catholic financial system in China, and the economic conflicts between the government and the Church (Kang 2019, pp. 3–4). Drawing from the *Contractuum collecta exemplaria* 契券匯式, Huang highlighted property disputes that commonly arose between the Catholic Church and Chinese society during the late Qing period. To assist missionaries in dealing with Chinese land and property transactions, Huang created juridical writing templates.

*Notions techniques sur la propriété en Chine* was a two-volume handbook that provided guidance for resolving property-related issues. The first volume contained legal and professional terminology commonly used in property transactions, while the second volume offered formatted templates for land leasing agreements and sales contracts. Originally published in Chinese in 1882, it was later translated into Latin by the T'ou-sè-wè Press. The North-China Branch of the Royal Asiatic Society, based in Shanghai, expressed interest and requested reprints in its affiliated journal (Shanghai Library 2013, pp. 118–43). This sparked academic attention, and Huang revised and republished the book in 1891, receiving praise for its practicality from Western scholars in China. A French translation, accomplished in 1897 by Jesuits Joseph Bastar 呂承望 and Jerónimo Tobar 管宜穆, appeared in the 11th monograph of *Variétés Sinologiques*, and was awarded the Prix Julien the following year.

Another individual who found value in Huang's book was Camilo Pessanha, a Portuguese Sinologist and poet residing in Macau. With a law degree from the University of Coimbra, Pessanha served as a judge in Macau and later worked at the Procuratura dos Negócios Sínicos (Zhang 2017, pp. 62–68). During his tenure, he encountered various dis-

putes concerning church property, and Huang's book proved to be a valuable resource for his administrative work. A rare edition of the book, bearing Pessanha's Chinese bookplate and the bookplate of the Procuratura dos Negócios Sínicos Library, is housed in the Macau Public Library, underscoring its significance as an essential document for official matters.

Literature Language textbooks of T'ou-sè-wè Press also played a significant role in promoting cross-cultural communication in various fields such as politics, economics, and law, integrating into the practical propositions of social sciences. These textbooks go beyond specialized purposes, enhancing the possibilities for versatile Chinese education and teaching. They strike a balance between practical value and adaptability, offering valuable experiences for adjustment based on real-world needs.

## 6. The Comparative Perspective in Scientific Textbooks

Modern Zi-ka-wei, known for its scientific and educational prominence, gained international recognition by establishing research institutions, attracting experienced academics, and implementing a distinct French-style scientific system. As an independent research unit, it relied on scientific textbooks to disseminate knowledge. These textbooks, categorized into mathematics, physics and natural sciences, and geography, not only drew from China's academic traditions, but also incorporated materials from the Jiangnan region, providing a comparative perspective. This process of compilation and organization played a vital role in fostering mutual understanding.

### 6.1. Publications in Humanistic Geography

Geography textbooks in Sinological research embraced a human geography perspective, aligning with the strategies employed in literature and language textbooks. These textbooks emphasized cross-language translation, professional certification, and flexible reorganization, effectively blending humanities with scientific research. The geography category consisted of 14 works.

Aloysius Richard 夏之時 made significant contributions, particularly with his highly acclaimed work, *Géographie de l'empire de Chine* 中國坤輿志略. This work garnered recognition from professional research societies such as the Société de Géographie and la Société de Géographie Commerciale, carrying on the French Jesuit tradition of geographical exploration. Richard's contribution was divided into two main parts: the China Proper section, which provided a general introduction to the eighteen provinces under direct central government administration, and sub-volumes that focused on northern, central, southern, coastal, and political–economic regions. The second part consisted of six volumes dedicated to Vassal States, exploring the tribal connections to the Qing court.

*Géographie de l'empire de Chine* remains a practical reference book, featuring appendices such as the Index of Places in the Eighteen Provinces, List of Chinese Provincial Governors, Index of Names of Cities and Towns, Index of Places Written in Chinese Characters, and Official Titles of Civil and Military Officials. The book also includes 51 maps, personally created by Aloysius Richard. In total, 15 maps are integrated into the text, while 35 are separate maps. The final map, a colorful fold-out version, was later printed separately as the *Sinica et gallica 18 provinciarum completa mappa* 十八省全圖.

The English translation of Richard's *Comprehensive Geography of the Chinese Empire* by Martin Kennelly 甘沛澍, an Irish Jesuit, expanded upon the concise French version. Spanning eight substantial volumes, bound in cloth and priced three times higher than the French edition, Kennelly's translation significantly supplemented and enriched the work. It served as a notable "byproduct" of this collective effort.

Henricus Dugout 屠恩烈's *Mappa Kiang-sou* 江蘇全圖, a comprehensive map continuously updated sheet by sheet from various locations, was included in the *Variétés Sinologiques* series as No. 54. It received recognition and awards from multiple societies, including la Société de Topographie.

During the late Qing and early Republican periods, the most significant geographical translation work was *Geographicorum Quinque Continentium* 五洲圖考. Larentio Li, as the

chief editor of *I-wen-lü* 益聞錄, dedicated a special world geography column to compile selections from this work. Chinese Jesuits, such as Simon Kiong 龔柴, Étienne Zi 徐勱, and Jean Baptise Hiu 許彬, took charge of different regions, providing detailed overviews accompanied by vivid descriptions of national history. This work gained widespread praise for striking a balance between scientific rigor and literary readability in geography textbooks. The *Catalogus* also listed Aloysius Van Hée 赫師慎's compilation and translation of *Concordantia sinicact europeea nominum geographicorum quinque continentium* 五洲地名中西合表, which facilitated cross-referencing.

Geography textbooks, influenced by the rise of natural history in Europe, gradually developed a knowledge system with global consciousness. They combined with travel writing (Pratt 1992, pp. 6–10) and marked the initial stage of introducing cultural geography data into China.

*6.2. The Influence of Pure French Ideals in the Compilation of Mathematics Textbooks*

The influence of pure French mathematical ideals can be observed in mathematics textbooks. In the *Catalogus*, only seven works in Pure Mathematics are listed, indicating a strong emphasis on the theoretical nature that is inherent in the French mathematical concept. The curriculum covers a wide range of topics, including both advanced mathematical research and elementary arithmetic, as defined by the New Educational System.

In the domain of advanced mathematics, the French mathematical system made its way into Sinological research through translations by the esteemed mathematician Carlo Bourlet. Two notable translations are *Geometria plana nova* 幾何學·平面 and *Algebra nova* 代數學, which were specifically designated for use at Aurora University and translated by Professor Lou Siang 陸翔. Professor Lou carefully selected a younger generation of French mathematicians as the basis for the translation, aiming to elucidate the French-style higher education philosophy at Aurora University, which received funding from the French Boxer Indemnities (Wang 1962, p. 377). Another significant translation was the last six volumes of *Elements* by Li Shanlan 李善蘭 and Alexander Wylie, further exemplifying the scientific methodology and preference for French mathematical ideals upheld by Jesuit institutions. The selection highlights the influence of the French mathematical system on Sinological research to convey the French-style higher education philosophy.

In the realm of elementary arithmetic, textbooks consistently adopted a question-and-answer format, influenced by the earlier works of German-origin French Jesuit Frank Scherer 佘賓王 at St. Ignatius College. These works, such as *Arithmetica in modum dialogi* 數理問答, *Geometria in modum dialogi* 量法問答, and *Algebra in modum dialogi* 代數問答, adhered to the advanced rules recognized by the French mathematical community. In contrast, Laurentio Li's translation of *Scientia clavis* 西學關鍵 focused on the core aspect of "numbers" and presented a vibrant interpretation with a philosophically conscious style. Through a lively dialogue format, it conveyed the idea that "Every discipline includes algorithms. For example, the measurement of heaven and earth, the analysis of chemical components, calculations related to the rise and fall of production in imports and exports—all of these can be examined with numbers." Furthermore, the original author of *Scientia clavis*, Aloysius Van Hée, was a prominent historian of arithmetic who explored renowned ancient Chinese mathematical problems such as the "Hundred Fowls Problem" 百雞問題 and the "Chinese Remainder Theorem" 剩餘定理. He published research on the achievements of ancient Chinese mathematics in the academic column of *T'oung Pao*, introducing them to the European Sinological community. However, due to a lack of alignment with French academic ideals, his work was classified under the "Miscellaneous" section.

*6.3. Integrating Jiangnan Data into the Global Scientific Network*

The Jesuit Jiangnan Scientific Plan played a vital role in updating and revising physics and natural science textbooks, reflecting the dynamic nature of scientific progress. This plan focused on geophysics and astronomy meteorology, organizing local data within the

framework of the "imported order". Zi-ka-wei Observatory served as a coordinate on the global knowledge map.

The establishment and maintenance of Zi-ka-wei Observatory were carried out by French Jesuit scientists Henry le Lec 劉德耀, Augustin Colombel 高龍鞶, and Marcus Dechevrens 能恩斯. Initially, the observatory was located on the banks of the Zhaojiabang Canal 肇嘉浜, where the scientists performed experiments within a few house terraces on basic living needs. They relied on limited instruments like meteorological recorders, thermometers, barometers, and anemometers for rudimentary observational activities.

In 1883, renowned French Jesuit Stanislas Chevalier 蔡尚質 arrived in Shanghai to set up instruments for the Xianxian Meteorological Observatory. When the Xianxian mission ended, the instruments were transported to Shanghai, providing the necessary equipment for astronomical observations and meteorological services in Zi-ka-wei. In the following year, Zi-ka-wei Observatory erected a 41 m-high anemometer tower and a signal tower on the Bund. These installations provided meteorological forecasting services for ships entering and leaving the Shanghai port on the Huangpu River and the East China Sea. The visual meteorological signal system developed at the observatory was later adopted by the Chinese Maritime Customs in various ports across China in 1898.

Zi-ka-wei Observatory had distinct scientific missions from the Zo-sé Observatory. It was one of the three main points used for the world's first longitude determination. Stanislas Chevalier surveyed the Yangtze River and determined the longitudes of more than fifty cities along the river, contributing to the further development and utilization of the Yangtze River. He also advocated for the establishment of the Shanghai Meteorological Society and reported progress in various departments through annual thematic papers. Stanislas Chevalier's contributions were widely acknowledged, and a biography was written about him (de Lapparent 1937). A major road in the Shanghai French Concession, Route Stanislas Chevalier, was named after him.

The column includes *De luna et species eius* 太陰圖說, which contains a French–Chinese combined version translated by Gao Jun 高均, a professor at Aurora University and an astronomical researcher in the newly-founded Academia Sinica (Gao 1987). The lunar photos in the book were taken by the Zo-sè Observatory and printed using the latest phototype technology.

Josephus Tardif de Moidrey 馬德賚, the leader of Loh-ka-pang Observatory, authored *Manuel de Météorologie* 氣學通詮 as a modern meteorological monograph. This work became an introductory textbook at Aurora University. The author continuously revised it to ensure that Aurora students could access cutting-edge achievements. The translated version by Liu Jinyu 劉晉鈺 and Pan Zhaobang 潘肇邦 was divided into four volumes, accompanied by numerous illustrations and five highly practical supplements, aligning modern French meteorology with traditional Chinese meteorological methods.

These achievements of the Jiangnan Scientific Plan not only influenced educational activities, but also facilitated the mutual nourishment of Chinese and Western scientific concepts and practices.

### 6.4. Science Textbooks: Unraveling Paradoxes and Cultural Exchange

The landscape of science textbooks reveals intriguing contradictions that warrant exploration. Firstly, these textbooks are primarily authored by Western scholars, albeit with contributions from Chinese scholars at Zi-ka-wei who aid in the translation process. As a result, these missionary-authored textbooks often incorporate Catholic doctrines, creating a perceived dichotomy between religion and science in both translation and teaching. Secondly, despite the recent surge in scientific translation activities and research, the science textbooks published by the T'ou-sè-wè Press receive little mention. This may be due to their limited readers, primarily consisting of a small number of professional researchers within the Zi-ka-wei institutions. However, these textbooks have garnered significant recognition from international academic societies, underscoring their importance within the global academic system. To unravel these puzzles, a comparative perspective is crucial.

For the Jesuits, the Intellectual Apostolate has long been a central component of their accommodation policy. Jesuit missionaries and academic collaborators at Zi-ka-wei have gradually shifted their research focus towards the established modern scientific order. They have diligently organized information about the Jiangnan region into established disciplinary systems, thus contributing to the global academic network. As the original authors of the scientific textbooks at Zi-ka-wei, they possess unparalleled expertise in their respective fields and embrace a tradition of tolerance that parallels the natural sciences. This tradition helps dissolve disciplinary barriers within the established system. For instance, Joseph Tardif de Moidrey's sinology studies were as significant as his contributions to metrology in works such as *Carte des Prefectures de Chine et de Leur Population Chrétienne en 1911* 中國各州府基督信眾分布圖 and *La Hiérarchie Catholique en Chine, en Corée au Japon, 1307–1914* 天主教在中國高麗日本六百年鐸階制度. These works are considered exemplary in the field of Chinese studies. Similarly, Joseph Chen 沈良 engaged in electrical engineering publications like *De electricitate* 實用電學, while also translating artistic works such as *Elements de perspective* 透視學撮要. Lou Siang, involved in translating mathematical textbooks, actively participated in Paul Eugène Pelliot's Dunhuang research and translated his monumental work *Une bibliothèque médiévale retrouvée au Kan-sou* 敦煌石室訪書記. Lou Siang also pursued his own historical research interests, completing *Chronology of the History of the Five Barbarians and Twenty States* 五胡二十國史表. Influenced by the tradition of the Shanghai School of Painting and Calligraphy, in which Lou's father was a leading figure, Lou Siang possessed excellent painting skills, which contributed to his work *Illustrations of Dunhuang* 敦煌圖錄. These individuals demonstrate the dynamic intersection of Dunhuang Studies and the unique intellectual environment of Zi-ka-wei, showcasing their multifaceted capabilities.

The scientific textbooks published by the T'ou-sè-wè Press conform to the academic order prevalent in Europe. Since the late Qing Dynasty, a wave of technological translation has brought numerous Western technical texts to the attention of Chinese scholars. Through the correspondence of Chinese and Western terminology, these texts aimed to clarify concepts and deepen Chinese understanding of Western learning. This led to a more optimistic and open attitude towards the social changes brought about by Western science and technology. Current research on scientific and technological translations primarily focuses on their "influence on modern China", examining the perspectives of the original authors, translators, recipients, and the interpretative processes related to technical concepts and scientific logic. In contrast, the scientific textbooks at Zi-ka-wei strive to establish mechanisms aligned with European academic standards. These mechanisms include operating networks for information exchange, building regular communication platforms for societies, ensuring frequent journal updates, and expanding the training of high-level research talents. Consequently, these textbooks exist on the periphery of mainstream narratives.

However, it is essential to emphasize that the comprehensive evaluation system driven by the scientific textbooks of the T'ou-sè-wè Press will reshape the educational concepts and scientific strengthening strategies within modern China's intellectual community. This transformative process requires a relatively long incubation period and patience, providing a novel lens to interpret the Jesuits' "adaptation strategy" and explore their navigation of the interplay between scientific and cultural aspects within the hierarchy of civilizations.

## 7. Conclusions

The Catholic missionaries who arrived in China in modern times, notably the Jesuits, pursued their mission with the accommodation strategy, distinct from Protestant missionaries. Their efforts spanned more than three centuries, demonstrating the enduring commitment of the Catholic Church in the country (Latorette 1929, p. 823). The Jesuits at Zi-ka-wei actively engaged in scientific, educational, and cultural activities amidst the turbulent 1930s, exemplified by the *Catalogus*. This publication reflected their astute consideration of local and universal factors, showcasing the transformative evolution of the T'ou-sè-wè



Press as a religious and cultural institution. The cataloging strategy adeptly navigated controversies, balanced institutional pressures, and proactively responded to social transformations and the delicate interplay of politics and religion. Through collaboration, the Jesuits produced influential educational works and integrated themselves into the international academic network, advocating for tolerance.

In 1936, the Vatican organized the Esposizione Mondiale della Stampa Cattolica, prompting a comprehensive survey of publications across Chinese dioceses. The *Catalogus* can be viewed as a preparatory precursor to this effort (Peng 2015). Therefore, the exemplary nature and far-reaching influence of this historical material call for thorough analyses and validations on a broader global stage, encompassing a comprehensive scientific framework that transcends disciplinary boundaries and extends into the realms of art, material culture, and beyond. By confirming and evaluating its significance through a multidisciplinary approach, we can gain a deeper understanding of the Catholic Church's mission in China and its profound impact on the intricate intersection of religion, culture, and society.

**Funding:** National Social Science Foundation of China: 21CZS048.

**Data Availability Statement:** No new data were created or analyzed in this study. Data sharing is not applicable to this article.

**Conflicts of Interest:** The author declares no conflict of interest. The funding sponsors had no role in the design of the study; in the collection, analyses, or interpretation of data; in the writing of the manuscript, and in the decision to publish the results.

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
