# Peer review of "Accommodation Strategies in Education: Exploring Jesuit Textbook Publication in Modern Shanghai"

_religions, doi:10.3390/rel15040385_

Round 1
Reviewer 1 Report
Comments and Suggestions for Authors
Compare foreign edited textbooks with local textbooks.
Explain the Jesuit education philosophy.
Compare Jesuits' textbooks in Shanghai with missionary textbooks in other cities or provinces
Comments on the Quality of English LanguageIt is fine, not much problem. Only minor refining
Author Response
Dear Reviewer,
I sincerely appreciate your time and effort in reviewing my work and your kind compliments. Your encouraging words serve as a great motivation for me to continue my research. I would also like to express my gratitude for providing insightful suggestions. Please find below my responses to your comments:
Comment 1: Comparisons are encouraged to be made between foreign-edited textbooks and local textbooks, as well as between Jesuits' textbooks in Shanghai and missionary textbooks in other cities or provinces.
Response 1: Thank you for the insightful suggestion. The influence of Western or Japanese edited textbooks on modern China was significant and extensive. In section "6.2. The Influence of Pure French Ideals in the Compilation of Mathematics Textbooks," I have chosen the example of the last six volumes of Elements by Li Shanlan and Alexander Wylie. Additionally, I will extend the conclusion to emphasize the impact of the French mathematical system on Sinological research in conveying the French-style higher education philosophy. I believe this revision will address your suggestion appropriately.
Comment 2: Explain the Jesuit education philosophy.
Response 2: In section "4.2.2. The Shared Philosophy," I have revised the paragraph to incorporate the Jesuit education philosophy as outlined in their official document, the Ratio. The revised paragraph now reads as follows:
"The establishment of the subject in modern Zi-ka-wei reflects the shared philosophy between the 1922 National Education Reform and the Jesuit Noveau Ratio Stadium (Pavur 2005, pp. 145-170). This approach recognizes the importance of adapting religious educational philosophy to the local context while also driving universality. In line with the Jesuit education philosophy found in the Ratio, there is a focus on academic specialization, particularly in science and theology, while also acknowledging the significance of Christian conduct and rhetoric (O'Malley 2000a). Modern Zi-ka-wei, such as St. Ignatius College, demonstrates this philosophy by making accommodation adjustments to its subject structure, exemplified by retaining the Course of Ethicæ alongside national education's Civics."
I trust that these revisions address your concerns and enhance the clarity and coherence of my work. Once again, I sincerely appreciate your valuable feedback, and I am committed to implementing your suggestions to improve the overall quality of my research.
Thank you for your time and consideration.
Sincerely,
Wei

Reviewer 2 Report
Comments and Suggestions for Authors
The author has done an excellent job describing the cross-cultural environment of Shanghai and Xujiahui during the early 20th century, focusing on a few publications directly related to Sino-European exchanges.
Despite some minor mistakes (e.g. p. 5 Parvae Narrationes and not Pavae Narrationes), the author has provided fascinating insights into the publication of books in Xuajiahui and Tushawan between the late 19th century and the early 20th century. In recent years, a growing number of scholars focused also on the material cultural productions in Xujiahui and Tushanwan ( e.g., William Ma (2016) The Art and Craft Workshops at the Catholic Orphanage of Shanghai (Tushanwan): French Jesuit’s Westernization, Proselytization, and Presentation of China (1863– 1937), Ph.D. dissertation, University of California, Berkeley) which should be at least mentioned in the paper. Moreover, there is more scholarship on specific Jesuit missionaries, like Fr. Angelo Zottoli which should be included in the paper (e.g. Antonio De Caro(2023), “Converting Zi-ka-wei: Fr. Angelo A. Zottoli (Chao Deli 晁德蒞,1826– 1902) and His Mission in Shanghai”, Journal of Jesuit Studies, 10, 640–653.; Liu Jinyu 刘津瑜, “晁德蒞、馬氏兄弟和拉丁文” (Zottoli, the Ma Brothers, and Latin), Wenhuibao xueren zhuankan 文匯報·學人專刊 (2015), 11.; Antonio De Caro, “Angelo Zottoli (1826–1902),” in Dizionario biografico degli italiani (Rome: Treccani, 2020), 100:804–7; Si Jia 司佳, “晁德蒞與清代 《聖諭廣訓》的拉丁文譯本” (Angelo Zottoli and the Latin translation of Amplification of the Sacred Edict in the Qing Dynasty), Fudan Journal (Social Sciences Edition) 58, no. 2 (2016): 65–72; Jaewon Ahn and Soojeong Moon, “Angelo Zottoli’s Observations on Enthymematic Features in Chinese Texts,” Rhetorica 38, no. 3 (2020): 309–20).
The article does not require major changes, but I would suggest providing further insights on those publications and their cross-cultural and intermedial relevance (many of them, for instance, also contained interesting prints on a wide variety of topics). This might include also a focus not only on scientific endeavors but also on art and material culture. What impact did prints have on the publications of those books? Was there any element combining Chinese and European cultures? What role did Chinese scholars play in the publication of those books and how did European Jesuit missionaries participate in this interaction? These questions might be discussed in the conclusion which, at the moment, would require a more in-depth analysis of the issues discussed in the article.
Overall, I would like to congratulate the author for this excellent article that, I am sure, will be of great interest to scholars in the field of Chinese Christianity and Sino-European exchanges. The article deserves to be published and it shows several important aspects of the Jesuit-led missions in Shanghai and their relevance during the late Qing dynasty and even beyond that period.
Comments on the Quality of English Language
The English is overall fine, with minor mistakes, but I would still recommend further polishing and proofreading. I also encourage the author to check carefully Latin and Chinese titles (including the Romanization into pinyin especially for the personal names of Chinese and European scholars), since sometimes there are inaccuracies or typos.
Author Response
Dear Reviewer,
Thank you for taking the time to read my work and for your kind compliments. Your words of encouragement greatly motivate me to continue with my research. I would also like to express my gratitude for providing me with insightful suggestions. Please find below my response to your comments:
Comment 1: Minor mistakes in the Latin Spelling.
Response 1: Thank you for pointing out the potential errors in the Latin spelling. I have thoroughly reviewed the Latin spelling of authors, books, and catalogs to ensure their accuracy.
Comment 2: Attention to the growing scholarships on Tushanwan and Fr. Zottoli.
Response 2: I appreciate the valuable references you have shared, including the works of De Caro and William Ma's contribution on the Tushanwan Atelier. I have incorporated these important references into the relevant sections of my work. Furthermore, I have expanded the Literary Review section to establish coherence and incorporate the additional references.
Comment 3: A more in-depth analysis on the cross-cultural and intermedial relevance in terms of art and material culture in the concluding remarks.
Response 3: In response to this suggestion, I have extended the concluding remarks to address the cross-cultural and intermedial relevance in terms of art and material culture. The revised conclusion emphasizes the need for a comprehensive scientific framework that transcends disciplinary boundaries. By adopting a multidisciplinary approach, we can thoroughly analyze and validate the significance of the historical material. This broader perspective allows for a deeper understanding of the Catholic Church's mission in China and its impact on the intricate intersection of religion, culture, and society.
Thank you once again for your valuable feedback. I have made the necessary revisions based on your suggestions, and I believe these enhancements have strengthened the overall quality and coherence of my work.
Best regards,
Wei Mo
